# Imaging Delay Following Liver-Directed Therapy Increases Progression Risk in Early- to Intermediate-Stage Hepatocellular Carcinoma

**DOI:** 10.3390/cancers16010212

**Published:** 2024-01-02

**Authors:** Jordin Stanneart, Kelley G. Nunez, Tyler Sandow, Juan Gimenez, Daniel Fort, Mina Hibino, Ari J. Cohen, Paul T. Thevenot

**Affiliations:** 1University of Queensland Medical School, Brisbane, QLD 4072, Australia; v-jstanneart@ochsner.org; 2Institute of Translational Research, Ochsner Health System, New Orleans, LA 70121, USA; kelley.nunez@ochsner.org (K.G.N.); mina.hibino@ochsner.org (M.H.); 3Interventional Radiology, Ochsner Health System, New Orleans, LA 70121, USA; tyler.sandow@ochsner.org (T.S.); juan.gimenez@ochsner.org (J.G.); 4Center for Applied Health Services Research, Ochsner Health System, New Orleans, LA 70121, USA; daniel.fort@ochsner.org; 5Multi-Organ Transplant Institute, Ochsner Health System, New Orleans, LA 70121, USA; acohen@ochsner.org; 6Faculty of Medicine, University of Queensland, Brisbane, QLD 4072, Australia

**Keywords:** hepatocellular carcinoma, liver directed therapy, liver transplantation

## Abstract

**Simple Summary:**

Hepatocellular carcinoma (HCC) remains one of the leading causes of cancer-related deaths worldwide. While there has been an improvement in detecting this cancer earlier on, progression rates have remained consistent. Patients with early-stage HCC are first treated with liver-directed therapies before they are eligible to undergo liver transplantation (LT). The success of liver-directed therapy (LDT) is measured by follow-up imaging and is crucial to the success of overall outcomes in HCC. In this study, we investigated the impact of HCC care delay, specifically delays in follow-up imaging studies, on tumor progression in patients with early-stage HCC. The results demonstrate a need to optimize the scheduling of post-treatment appointments to decrease HCC care delay and improve progression rates.

**Abstract:**

Hepatocellular carcinoma (HCC) remains one of the leading causes of cancer-related deaths in the world. Patients with early-stage HCC are treated with liver-directed therapies to bridge or downstage for liver transplantation (LT). In this study, the impact of HCC care delay on HCC progression among early-stage patients was investigated. Early-stage HCC patients undergoing their first cycle of liver-directed therapy (LDT) for bridge/downstaging to LT between 04/2016 and 04/2022 were retrospectively analyzed. Baseline variables were analyzed for risk of disease progression and time to progression (TTP). HCC care delay was determined by the number of rescheduled appointments related to HCC care. The study cohort consisted of 316 patients who received first-cycle LDT. The HCC care no-show rate was associated with TTP (*p* = 0.004), while the overall no-show rate was not (*p* = 0.242). The HCC care no-show rate and HCC care delay were further expanded as no-show rates and rescheduled appointments for imaging, laboratory, and office visits, respectively. More than 60% of patients experienced HCC care delay for imaging and laboratory appointments compared to just 8% for office visits. Multivariate analysis revealed that HCC-specific no-show rates and HCC care delay for imaging (*p* < 0.001) were both independently associated with TTP, highlighting the importance of minimizing delays in early-stage HCC imaging surveillance to reduce disease progression risk.

## 1. Introduction

Primary liver cancer is the third-leading cause of cancer-related deaths worldwide, with over 900,000 new cases diagnosed globally in 2020 [1]. Hepatocellular carcinoma (HCC) accounts for over 75% of all primary liver cancer cases [1]. Despite continued advancements in HCC surveillance and early-stage diagnosis, HCC-related mortality remains high [2], with a 2.1% increase in mortality from 1999 to 2016 in the United States [3]. HCC is frequently preceded by liver cirrhosis, which may occur secondary to various etiologies, such as viral infection, alcohol abuse, and metabolic syndrome [4]. While surgical resection is the mainstay treatment for resectable HCC, liver transplantation (LT) remains the only curative treatment for nonresectable HCC [5].

Liver transplant candidates with nonresectable HCC are subject to a mandated 6-month waiting period prior to being granted exception points that increase priority status on the transplant waiting list [6]. As a result, liver-directed therapy [7] has become standard of care during this waiting period to delay disease progression and bridge/downstage patients to LT [5]. Liver-directed therapy (LDT) has also been used as a definitive treatment option for early-stage HCC [8,9]. The response to LDT has been shown to be an important indicator as a bridge to LT success as well as risk of post-LT recurrence [5]. Assessing this response is dependent on timely imaging; it is important to look at what factors may impact HCC care and how it can be optimized. 

Healthcare delay impacts HCC patients at multiple points throughout their treatment course. Significant delays in HCC screening have been shown amongst patients with known cirrhosis [10]. While screening programs have improved early-stage HCC diagnosis, one recent study including all stages of HCC demonstrated that 13% of patients experienced a diagnostic delay of >3 months from initial presentation to diagnosis [11]. Moreover, treatment delay across all HCC-specific treatment modalities (resection, LDT, transplantation, and systemic therapy) has been shown to significantly impact survival outcomes in early- to advanced-stage HCC [7,11]. It was further demonstrated that these treatment delays were driven by racial and socioeconomic disparities [7]. These studies focused on treatment delays from HCC diagnosis to surgical or therapeutic treatments. In this study, we sought to evaluate the role of HCC care delays along a longer timeframe, from diagnosis to progression, and understand its impact on TTP in nonresectable HCC being bridged/downstaged to liver transplantation.

## 2. Materials and Methods

### 2.1. Study Design and Patient Population

This single-center cohort study was created longitudinally by reviewing the electronic medical record database using Epic Slicer Dicer software to identify all patient encounters for LDT within the Ochsner Health System. Extracted records were screened for study inclusion criteria: (i) HCC diagnosis confirmed by biopsy or triple-phase imaging in accordance with Liver Reporting & Data System criteria; (ii) nonresectable HCC; (iii) treated with LDT as a bridge to LT, downstage to LT, or definitive treatment plan; (iv) Child Pugh A-B; (v) Barcelona Clinic Liver Cancer (BCLC) A-B; (vi) Eastern Cooperative Oncology Group (ECOG) 0–1; and (vii) within Milan Criteria. Exclusion criteria included: (i) co-malignancies, (ii) race other than Caucasian or African American, (iii) out-of-state residency, (iv) lack of healthcare insurance, and (v) missing demographic information from electronic medical record. This study was approved by the Ochsner Health System Institutional Review Board (protocol 2019.308) and conducted in accordance with the ethical principles set forth by the Declarations of Helsinki.

### 2.2. Study Variables

General demographics, cirrhosis history and serology, and HCC diagnostic baseline variables were extracted from the medical record at the time of HCC diagnosis. Serology and multidisciplinary HCC tumor board reports were used to compute baseline cirrhosis (Child Pugh) and HCC staging (Milan Criteria, ECOG performance status, and BCLC staging). Distance from center was determined using freemaptools.com to find the direct point-to-point distance between the patients’ residence zip code and the transplant center zip code. Liver transplantation track status was determined based on whether the patient underwent liver transplant evaluation with transplant coordinators, as recommended by hepatologists and noted in their EMR.

### 2.3. Care Delay Variables

Overall no-show rate was calculated as a percentage of all no-show appointments over total appointments within the healthcare system. No-show and rescheduled appointments specific to HCC care were defined by isolating orders originating from hepatology, interventional radiology, or transplant hepatology occurring (i) after the HCC diagnosis date and (ii) prior to the primary endpoint or censored outcome date. Office visits included both in-person clinic and telehealth appointments. HCC care no-show rate was calculated as the number of no-show appointments (imaging, laboratory, and office) over the duration of time on study from first-cycle LDT date or until censored endpoint in months. HCC care no-show rate was broken down further to number of no-show appointments for imaging (HCC imaging no-show rate), laboratory (HCC laboratory no-show rate), or office (HCC office no-show rate) encounters. HCC care delay consisted of the amount of rescheduled imaging (HCC imaging delay), laboratory (HCC laboratory delay), and office (HCC office delay) over the duration of time on study, as defined for HCC care no-show rate. Delays after first-cycle LDT for imaging, laboratory, or office appointments were each calculated as the time delay over the returned to clinic order provided by the interventional radiologist. 

### 2.4. Liver-Directed Therapy Sites and Treatment Protocols

Liver-directed therapy was performed at interventional oncology sites within Louisiana (Baton Rouge, New Orleans, Shreveport) within the referral network of a single liver transplant center (Ochsner Health Multi-Organ Transplant Institute, New Orleans, Louisiana, USA). The multidisciplinary HCC board includes transplant hepatologists, transplant surgeons, interventional radiologists, and oncologists and assesses patients from all interventional oncology sites. Institutional criteria for LDT as a bridge/downstage/definitive treatment: (i) BCLC A-B, (ii) Child Pugh A-B, (iii) nonresectable HCC, (iv) without main portal vein thrombus or extrahepatic metastasis, (v) total bilirubin < 4 mg/dL, (vi) serum creatinine concentration < 1.5 mg/dL, and (vii) absent gross ascites by ultrasound or CT. The treatment approach and modality are determined by the multidisciplinary HCC board and are dependent upon patient-specific variables such as index tumor size and location. Treatment modalities included drug-eluting embolic transarterial chemoembolization (DEE-TACE), ^90^Yttrium transarterial radioembolization (^90^Y), and percutaneous microwave ablation (MWA). The institutional treatment algorithm utilized MWA for an ablatable index HCC ≤ 3 cm and ^90^Y for nonablatable index HCC > 3 cm as well as multifocal disease. Patients with contraindications to both MWA and ^90^Y received DEE-TACE.

DEE-TACE was performed using 100–300 μm beads (LC Bead, BTG, London, UK) containing 50 mg/vial doxorubicin. The two-phage ^90^Y procedure included pre-treatment mapping angiogram with ^90^Y glass microsphere infusion occurring 2–4 weeks later (TheraSphere; Boston Scientific, Marlborough, MA, USA). All tumors were treated with a target radiation dose greater than 200 Gy. Percutaneous MWA was performed using a high-powered, gas-cooled, multiple antenna system (Neuwave Medical, Madison, WI, USA). The performing physician dictated the duration of treatment and power application with respect to manufacturer guidelines and modifications considered for tumor size and/or proximity to vulnerable structures. The ablative margin was set at >5 mm for all MWA procedures. 

Objective response rate (ORR) was recorded using the Response Evaluation Criteria in Solid Tumors modified for HCC (mRECIST) [12] using follow-up imaging to assess for first-cycle response to LDT. The timeline for follow-up imaging was modality-dependent and targeted for 30 days (DEE-TACE and MWA) or 60–90 days (^90^Y). Patients with a predetermined treatment plan consisting of multiple cycles of LDT to the index lesion had follow-up imaging and response evaluated following the final treatment in the sequence. The multidisciplinary HCC board determined ongoing treatment plans based on the follow-up imaging. Patients with residual disease received additional treatment cycles until progression to BCLC-C. Patients with a satisfactory response to treatment were followed over recurring 3-month surveillance intervals until definitive tumor response, surgical intervention, or the need for additional treatment for BCLC A-B disease.

### 2.5. Primary Outcome

The primary outcome for this study was HCC progression beyond transplant criteria. HCC progression was defined as no longer being a transplant candidate due to progression beyond Milan Criteria or failure to downstage within Milan Criteria, as determined by the multidisciplinary HCC board based on follow-up or surveillance triple-phase imaging. The assessment of primary outcomes was performed using logistic regression and time-to-progression analyses. Analysis looked solely at patients who either received LT or had HCC progression. Reasons for censoring included: (i) patients electing to pursue systemic therapy without tumor progression, (ii) lost to follow-up (>6 months), (iii) all-cause mortality, or (iv) active in treatment without progression or having received LT at the time of data analysis.

### 2.6. Statistical Analysis

Data analysis was performed in JMP 16.0 (SAS Institute Inc., Cary, NC, USA), and graphs were generated using Prism 9.4.1 (GraphPad Software Inc., Boston, MA, USA). Categorical variables were reported as number and percentage of cohort, and continuous variables were reported as median with interquartile range (IQR). Univariate and multivariate analyses of the factors associated with TTP were performed using the Cox proportional hazards model. Components of established clinical indices for HCC progression risk were excluded from multivariate analysis despite reaching significance in the univariate analysis in favor of the clinical indices of interest. The mRECIST score could not be assessed in patients with missing follow-up data and, thus, they were excluded from univariate analysis.

## 3. Results

### 3.1. Cohort Demographics

The study cohort consisted of 316 HCC patients undergoing first-cycle LDT to bridge/downstage to LT or definitive treatment. General demographics, baseline hepatology, HCC characteristics, and response to treatment are summarized in the *Data* column in Table 1. The median age was 63 years, 75% (238/316) were male, and they were predominately Caucasian (222/316, 70%). Primary cirrhosis etiology was mostly hepatitis C virus (57%, 181/316), with 69% (217/316) of patients having Child Pugh A. The median index lesion size was 2.8 cm, with 93% (294/316) having HCC staging of BCLC A and median alpha fetoprotein (AFP) levels of 13.2 ng/mL. First-cycle LDT consisted of DEE-TACE (40%, 125/316), ^90^Y (37%, 117/316), or MWA (23%, 74/316), with 73% (212/316) of the cohort having an ORR to LDT. The median time from HCC diagnosis to first-cycle LDT was 56 days (IQR: 37–83), with a median time from first-cycle LDT to follow-up imaging of 42 days (IQR: 32–75). Most of the cohort was under evaluation for liver transplantation (82%, 259/316) at the time of LDT.

Overall, 42% (134/316) of the cohort was placed on the transplant waitlist. At the time of analysis, 32% (102/316) of the cohort was successfully bridged to transplantation, 22% (70/316) were still active in the study, 21% (66/316) were censored, and 25% (78/316) experienced HCC progression precluding access to LT (Table A1). The median follow-up time for the cohort was 10 months (IQR: 5–19).

### 3.2. Baseline Factors Associated with Time to Progression

Univariate analysis was performed to determine the factors associated with TTP following LDT (*p*-value column, Table 1). The median TTP for the cohort following first-cycle LDT was 38 months. Univariate analysis revealed that pre-treatment albumin and INR along with well-characterized risk factors, including tumor burden (Milan criteria, index lesion size, and BCLC staging), AFP levels, and response to LDT, were all associated with TTP. The HCC care no-show rate also emerged as a factor associated with TTP, while the overall no-show rate, race, insurance, or transplant track status did not.

### 3.3. Role of HCC Care Delay on Time to Progression

The significance of the HCC care no-show rate on TTP in the univariate analysis prompted further investigation into which types of HCC-specific care appointments were impacting progression risk. HCC care was broken down further into HCC care no-show rates (no-show appointments) and HCC care delays (rescheduled appointments), as shown in Table 2. In the cohort, 60% (190/316) of patients had at least one HCC imaging no-show appointment compared to 44% (142/316) for office visits related to HCC care. The number of patients with ≥3 no-show appointments was similar for imaging (19%, 62/316), laboratory (18%, 56/316), and office visits (13%, 41/316) related to HCC care. There was a large discrepancy in the number of rescheduled office appointments compared to imaging following LDT. Only 8% (26/316) of the cohort had ≥3 rescheduled office appointments compared to 61% (192/316) of rescheduled imaging appointments. The HCC no-show rate and care delay (rescheduled appointments) were normalized over follow-up time to control for the differences in length of treatment timelines between patients. Univariate analysis revealed that the HCC imaging no-show rate, HCC imaging, laboratory, and office delay were all associated with TTP (Table 3). Time to first-cycle treatment and time to first-cycle follow-up did not reach significance for TTP (*p* = 0.051). 

Multivariate regression controlling for Milan criteria with significant factors from the univariate analysis was performed. The HCC imaging no-show rate and HCC imaging delay were shown to be independent risk factors for TTP when accounting for Milan criteria, AFP levels, albumin levels, and first-cycle LDT response rates (Table 4).

### 3.4. Transplant Track vs. Non-Transplant Track Status on Time to Progression

To control for increased clinical engagement associated with the liver transplant evaluation process, patients were divided into transplant and non-transplant tracks based on referral for transplant evaluation. Most patients were under transplant evaluation (259/316, 82%) prior to first-cycle LDT. Patients who did not meet the transplant center’s criteria were deemed on the non-transplant track due to co-morbidities and social issues including substance abuse and lack of social support. Logistic regression analysis was used to reexamine the baseline characteristics including HCC care no-show rates and delays based on transplant track status (Table 5). Age, bilirubin, and AFP level at the time of LDT were shown to be significantly different between the two groups. Importantly, there were no significant differences in the HCC care no-show rate, HCC imaging rate, or imaging delay based on transplant track status.

Multivariate analysis showed that the HCC imaging no-show rate and HCC imaging delay remained independent risk factors for TTP when accounting for Milan criteria, AFP levels, albumin levels, and first-cycle LDT response rates, regardless of transplant track status (Table 6). 

## 4. Discussion

There has been a renewed focus on the role of socioeconomic factors in HCC outcomes, with a particular focus on access/adherence to disease surveillance and overall survival outcomes [13,14,15,16]. The importance and optimal application of post-treatment imaging schedules in early- to intermediate-stage HCC have been extensively reported in the literature and included in practice guidelines [17]. Unfortunately, socioeconomic factors may present a significant barrier to 1–3-month post-LDT surveillance and 3–6-month ongoing surveillance for disease recurrence. In this multi-center study within a single health system, a relationship between the overall no-show rate and risk of HCC progression was further interrogated to identify post-LDT, HCC care-specific imaging no-show/reschedule rates as a risk factor for HCC progression. The risk associated with post-LDT surveillance was maintained after controlling for disease management track, baseline HCC burden and biological aggressiveness, and initial tumor response rate.

Despite advancements in HCC surveillance protocols resulting in earlier detection of HCC [18], failure to deliver timely HCC-specific treatments continue to impact outcomes [7,19]. As these early detection protocols have reduced the prevalence of diagnostic delays [11], research has since largely shifted focus towards the impacts of treatment delay across all HCC treatment modalities [7,19,20]. Studies have demonstrated the impact of treatment delay on overall survival but have combined several treatment modalities (surgical resection, LDT, external radiation, liver transplantation, and systemic therapy) across multiple HCC stages from early to advanced [7,11,19,20]. Studies investigating HCC care delay and its impact on tumor progression risk in a uniform population undergoing a single treatment option are lacking. In this study, the impact of HCC care delays on TTP was investigated specifically in nonresectable HCC receiving LDT as a bridge/downstage to LT or as definitive treatment. Within this specific patient population, imaging delays in the form of no-show and rescheduled appointments were found to be independently associated with TTP risk.

Response to LDT is based on imaging [12] and has been shown to impact bridge to LT success [5,21]. Nonresponse to LDT has been shown to increase post-LT recurrence rates [21,22] with viable tumor on explant [21]. Guidelines for assessing the response following LDT are dependent on modality with imaging typically performed 30 days following MWA and DEE-TACE and 60–90 days following ^90^Y [23], though these timelines may differ from institution to institution. In this study, the median times for imaging following first-cycle MWA and DEE-TACE were 35 days (IQR: 31–48), and first-cycle ^90^Y was 75 days (IQR: 43–94). Although the median time from first-cycle LDT to first-cycle follow-up imaging approached significance (*p* = 0.051), and the greatest impact on TTP was HCC imaging delay. This includes imaging delays throughout a patient’s HCC care timeline during the waitlist for the LT window. Our results demonstrate the impact of an accumulation of HCC care delay focused on obtaining imaging following first-cycle LDT and during surveillance. Most of the patients on a bridge/downstage to LT timeline undergo more than one treatment. Nonresponders to LDT will often undergo additional LDT to continue to treat tumor burden. Within the cohort, 54% (170/316) of patients received additional LDT, of which 61% (104/170) experienced ≥3 rescheduled imaging appointments. Despite median times falling within the range, 61% (195/316) of the cohort experienced delays in HCC imaging. Delays in imaging will greatly impact the ability to assess how the tumor has responded to LDT and further delay transplant eligibility and greatly impact treatments still available. 

The frequency of office visit delays was substantially lower than follow-up/surveillance imaging delay. It is worth noting that the imaging appointments are likely to be the costliest and most time-consuming visits compared to the other follow-up requirements. Not to mention, due to the special nature of the diagnostic imaging requirements, these visits are more geographically limited as well. These barriers may explain the higher no-show prevalence among imaging visits compared to the other visit types. An additional explanation for this discrepancy could be due to the use of telehealth in the post-COVID-19 era. One study has investigated that no-show rates for telehealth visits were significantly lower than face-to-face visits for psychiatric care [24]. Although telehealth may reduce care gaps, a recent study comparing face-to-face and telehealth in HCC found similar rates of HCC diagnosis, hospitalizations, and mortality [25]. While the number of telehealth visits was not assessed, the use of telehealth for office visits is actively practiced in our hospital system in the post-COVID-19 era to provide greater access to healthcare. What percentage of office visits were conducted via telehealth was not monitored and warrants further investigation. One solution to decrease delays in HCC imaging could be to schedule post-treatment and surveillance imaging, laboratory, and office visits in succession on the same day. This may lessen the impact of transportation and time away from work or caring for family.

At transplant centers, the transplant care team made of hepatologists, transplant coordinators, social workers, case managers, and transplant surgeons is assigned to each patient to assist in the process of being placed on the transplant waitlist in a timely manner. The cohort was grouped based on referral for LT evaluation to control for additional clinical management and engagement and did not impact TTP. While a majority of the cohort was referred for LT evaluation (82%, 259/316), the 18% of the cohort that were not referred represent a portion of the population that may be at great risk for HCC care delay due to social issues. Notably, the geographic location of the transplant center in this study provides a heterogenous population sample that includes patients living in areas of both affluent and disadvantaged statuses [26,27]. Taken together, this provides a unique opportunity to examine HCC care delay after diagnosis and amongst a socioeconomically diverse population. There were also no significant differences in race between patients being evaluated for transplant versus those on a non-transplant treatment track. Additionally, it is well established that health insurance status is another barrier to receiving a liver transplant; particularly, patients without insurance or with Medicaid coverage have both delayed treatment and lower rates of treatment [28,29]. This patient population also exhibits a higher risk of death [29,30]. In this study, insurance type at the time of diagnosis did not impact TTP; however, further investigation into socioeconomic disparities and their impact on HCC care delay following diagnosis is warranted.

To fully appreciate the clinical significance of these findings, HCC care delay must be further interrogated to identify (i) how imaging no-show/rescheduling translates to a specific amount of time off imaging protocol, and (ii) what socioeconomic factors are driving care delay. A preliminary analysis of delay time in this cohort suggests that delays result in up to 1-month median variance from protocol for each delay event, with even greater cumulative delay in patients with multiple delayed imaging encounters. Perhaps even more concerning, in most cases, the contributing socioeconomic factor associated with the delayed encounter is unknown. It will be critical to utilize digital healthcare tools to identify these unknown socioeconomic factors associated with care delay and develop strategies to minimize care delay to improve early- to intermediate-stage HCC outcomes.

While this study has several strengths including a large, uniform cohort for nonresectable HCC undergoing LDT with a focus on HCC care delay, there are several limitations. This study is based in a single center with a uniform LDT treatment algorithm and bridge/downstage to LT protocol that may differ at other institutions. There was also a shift in the institution’s LDT treatment algorithm, which primarily utilized DEE-TACE from 2016 to 2020, then switched to ^90^Y and MWA (2020–2022). However, the purpose of the study was focused on investigating HCC care delay in a bridge/downstage to LT cohort in which all patients underwent at least one LDT treatment. Another limitation is our definition of transplant and non-transplant tracks. Transplant track was defined as patients who underwent transplant evaluation, not based on active transplant waitlist status. The referral process for liver transplant eligibility occurs within the first 1–2 months after HCC diagnosis. Patients that presented with social issues, substance abuse, and co-morbidities were not referred for evaluation. The purpose behind this grouping was to determine whether those reasons were driving HCC care delay. However, this did not account for patients that did not complete transplant evaluation for the same reasons listed above as referral for transplant evaluation was dependent on hepatologist’s referral. Additional work is warranted to further investigate the transplant evaluation process and its potential impact on HCC care delay.

## 5. Conclusions

In conclusion, HCC imaging delay is directly associated with disease progression risk in bridge/downstage to LT or definitive treatment in nonresectable HCC following LDT. The increased risk associated with care delay was effectively controlled for disease burden, biological aggressiveness, and response rate, suggesting a tumor-independent underlying factor contributing to care delay. To reduce disease progression risk, the critical socioeconomic factors contributing to HCC care delay must be identified where they can potentially be addressed by improving patient education, care coordination, and access to support resources.

## Figures and Tables

**Table 1 cancers-16-00212-t001:** Cohort demographics and time-to-progression univariate analysis.

Parameters	Data	Univariate Variable Input	*p*-Value
Cohort, *n*	316		
Study period, date range	4/21/16–4/6/22		
General Demographics			
Age at diagnosis, median (IQR)	63 (60–66)	Per 1 year change	0.671
Legal sex, *n* male (%)	238 (75)	Male vs. Female	0.731
Race, *n* (%)		Caucasian vs. African American	0.591
Caucasian/White	222 (70)		
African American/Black	94 (30)		
Insurance type, *n* (%)		Government vs. Private	0.755
Government	250 (79)		
Private	66 (21)		
Insurance type, *n* (%)		Medicaid vs. Other	0.255
Medicaid	51 (16)		
Other	265 (84)		
Distance from center in miles, median (IQR)	47.9 (12.9–84.3)	Per 1 mile change	0.420
Lives in state, *n* (%)	257 (81)	In state vs. Out of state	0.152
Overall No-Show Rate, %, median (IQR)	6 (3–12)	Per 1% change	0.242
HCC Care No-Show Rate/Month, median (IQR)	0.23 (0.08–0.53)	Per 0.1 change	0.004
Hepatology at Diagnosis			
Cirrhosis etiology, *n* (%)		HCV vs. NASH vs. ACV + ALD vs. ALD vs. Other	0.924
HCV	181 (57)		
NASH	42 (13)		
HCV + ALD	40 (13)		
ALD	27 (9)		
Other	26 (8)		
Child Pugh, *n* (%)		A vs. B	0.092
A	217 (69)		
B	99 (31)		
History of Decompensation, *n* (%)	89 (28)	Yes vs. No	0.446
Sodium mM, median (IQR)	139 (137–141)	Per 1 mM change	0.451
Creatinine mg/dL, median (IQR)	0.9 (0.8–1.1)	Per 1 mg/dL change	0.15
Bilirubin mg/dL, median (IQR)	1 (0.6–1.5)	Per 1 mg/dL change	0.108
Albumin g/dL, median (IQR)	3.4 (3–3.7)	Per 1 g/dL change	0.002
Albumin g/dL, *n* (%)		≤3.4 vs. >3.4	0.041
Albumin ≤ 3.4			
Albumin > 3.4			
INR, median (IQR)	1.1 (1–1.2)	Per 0.1 change	0.039
MELD-Na, median (IQR)	9 (7–11)	Per 1 score change	0.449
MELD 3.0, median (IQR)	10 (8–12)	Per 1 score change	0.159
HCC Baseline			
Multifocal, *n* (%)	60 (19)	Yes vs. No	<0.001
Index lesion cm, median (IQR)	2.8 (2.2–3.6)	Per 0.1 cm change	0.002
Index lesion cm, *n* (%)		Small vs. Intermediate/Large	0.013
Small (<3 cm)	190 (60)		
Intermediate/Large (>3 cm)	126 (40)		
Milan Criteria, *n* (%)	277 (88)	Yes vs. No	0.002
BCLC Stage, *n* (%)		A vs. B	0.002
A	294 (93)		
B	22 (7)		
ECOG, *n* (%)		0 vs. 1	0.663
0	205 (65)		
1	111 (35)		
Transplant Track, *n* (%)		In Evaluation vs. Not Under Evaluation	0.234
In Evaluation	259 (82)		
Not Under Evaluation	57 (18)		
AFP ng/mL, median (IQR)	13.2 (5.2–66.3)	Per 1 ng/mL change	0.046
AFP ng/mL, *n* (%)		<20 vs. >20	<0.001
AFP < 20 ng/mL, *n* (%)	182 (58)		
AFP > 20 ng/mL, *n* (%)	132 (42)		
Index Liver-Directed Therapy			
Modality, *n* (%)		DEE-TACE vs. ^90^Y vs. MWA	0.280
DEE-TACE	125 (40)		
^90^Y	117 (37)		
MWA	74 (23)		
Time from HCC Diagnosis to LDT, median (IQR)	56 (34–83)	Per 1 day change	0.926
Time from LDT to 1′ Follow up, median (IQR)	42 (32–75)	Per 1 day change	0.672
First-Cycle Response, *n* (%)		ORR vs. NOR	<0.001
ORR	212 (73)		
NOR	80 (27)		

Abbreviations: Interquartile range (IQR), Hepatocellular carcinoma (HCC), Hepatitis C virus (HCV), Nonalcoholic steatohepatitis (NASH), Alcoholic liver disease (ALD), International normalized ratio (INR), Model for end-stage liver disease (MELD); Barcelona Clinic Liver Cancer (BCLC), Eastern Cooperative Oncology Group (ECOG), Alpha-fetoprotein (AFP), Doxorubicin eluting embolic transarterial chemoembolization (DEE-TACE), Yttrium-90 (^90^Y), Microwave ablation (MWA), Liver-directed therapy (LDT), Objective response rate (ORR), Non-objective response rate (NOR).

**Table 2 cancers-16-00212-t002:** HCC care delay in no-show and reschedule appointments.

	Appointments		
HCC Care Delay	None	1–2	≥3
No-Show Appts for HCC Imaging, *n* (%)	126 (40)	128 (41)	62 (19)
No-Show Appts for HCC Labs, *n* (%)	162 (51)	98 (31)	56 (18)
No-Show for HCC Office Visits, *n* (%)	175 (55)	101 (32)	41 (13)
No. of Rescheduled HCC Imaging, *n* (%)	39 (12)	85 (27)	192 (61)
No. of Rescheduled HCC Labs, *n* (%)	47 (15)	64 (20)	205 (65)
No. of Rescheduled HCC Office, *n* (%)	211 (67)	79 (25)	26 (8)

Abbreviations: Hepatocellular carcinoma (HCC), Appointments (Appts).

**Table 3 cancers-16-00212-t003:** Univariate analysis of HCC care delay breakdown with time to progression.

HCC Care No-Show Rate	Data	*p*-Value	HR (95% CI)
Total No. of No-Show Appts for HCC Imaging	1 (0–2)		
HCC Imaging No-Show Rate/Month, median (IQR)	0.07 (0–0.22)	<0.001	11 (4.8–23.5)
Total No. of No-Show Appts for HCC Labs	0 (0–2)		
HCC Laboratory No-Show Rate/Month, median (IQR)	0.0 (0.0–0.16)	0.648	
Total No. of No-Show for HCC Office Visits	0 (0–1)		
HCC Office No-Show Rate/Month, median (IQR)	0.0 (0.0–0.11)	0.222	
HCC Care Delay			
Total No. of Rescheduled HCC Imaging	3 (1–7)		
HCC Imaging Delay/Month, median (IQR)	0.33 (0.10–1.0)	<0.001	1.8 (1.5–2.2)
Total No. of Rescheduled HCC Labs	4 (1–8)		
HCC Laboratory Delay/Month, median (IQR)	0.33 (0.11–1.0)	<0.001	1.6 (1.4–1.8)
Total No. of Rescheduled HCC Office	5 (2–9)		
HCC Office Delay/Month, median (IQR)	0.49 (0.19–1.2)	<0.001	1.8 (1.6–2.2)
HCC Imaging Delay after First-Cycle LDT, months, median (IQR)	0.53 (0.25–1.0)	0.051	
HCC Lab Delay after First-Cycle LDT, months, median (IQR)	0.77 (0.27–1.4)	0.840	
HCC Office Visit Delay after First-Cycle LDT, months, median (IQR)	0.80 (0.50–1.2)	0.760	

Abbreviations: Hepatocellular carcinoma (HCC), Hazard ratio (HR), Confidence Interval (CI), Appointments (Appts), Interquartile range (IQR), Liver-directed Therapy (LDT).

**Table 4 cancers-16-00212-t004:** Time-to-progression multivariate analysis.

	Univariate		Milan Multivariate Model	
Parameters	*p*-Value	HR (95% CI)	*p*-Value	HR (95% CI)
HCC Imaging No-Show Rate/Month	<0.001	11 (4.8–23.5)	<0.001	5.6 (2.6–11.6)
HCC Imaging Delay/Month	<0.001	1.8 (1.5–2.2)	<0.001	1.7 (1.4–2.1)
HCC Laboratory Delay/Month	<0.001	1.6 (1.4–1.8)	0.169	
HCC Office Delay/Month	<0.001	1.8 (1.6–2.2)	0.683	
Albumin, g/dL				
≤3.4 vs. >3.4	0.041	1.6 (1.0–2.5)	0.028	1.7 (1.0–2.7)
AFP, ng/mL				
>20 vs. <20	<0.001	3.0 (1.9–4.9)	<0.001	2.3 (1.4–3.7)
Milan Criteria				
Outside vs. Within	0.002	2.5 (1.5–4.3)	0.261	
First-cycle LDT Response				
NOR vs. ORR	<0.001	5.8 (3.7–9.2)	<0.001	4.1 (2.6–6.7)

Abbreviations: Hazard ratio (HR), Confidence Interval (CI), Hepatocellular Carcinoma (HCC), Alpha-fetoprotein (AFP), Liver-directed therapy (LDT), Objective response rate (ORR), Non-objective response rate (NOR).

**Table 5 cancers-16-00212-t005:** Transplant track vs. non-transplant track.

Parameters	Under Evaluation	Not Under Evaluation	*p*-Value
Cohort, *n*	259	57	
Study period, date range	4/21/16–3/10/22	4/25/16–4/6/22	
General Demographics			
Age at diagnosis, median (IQR)	63 (59–66)	65 (62–69)	0.001
Legal sex, *n* male (%)	192 (74)	46 (81)	0.287
Race, *n* (%)			0.516
Caucasian/White	184 (71)	38 (67)	
African American/Black	75 (29)	19 (33)	
Insurance type, *n* (%)			0.973
Government	205 (79)	45 (79)	
Private	54 (21)	12 (21)	
Distance from center in miles, median (IQR)	47.5 (13–89)	48.32 (12.44–81.79)	0.699
Lives in state, *n* (%)	216 (83)	51 (72)	0.054
Overall No-Show Rate, %, median (IQR)	6 (3–11)	7 (4–13)	0.161
HCC Care No-Show Rate/Month, median (IQR)	0.23 (0.07–0.53)	0.25 (0.09–0.58)	0.630
HCC Imaging No-Show Rate/Month	0.06 (0.0–0.22)	0.08 (0.0–0.20)	0.320
HCC Imaging Delay/Month	0.32 (0.10–0.89)	0.33 (0.11–0.33)	0.826
Hepatology at Diagnosis			
Cirrhosis etiology, *n* (%)			
HCV	150 (58)	31 (54)	0.313
NASH	38 (15)	4 (7)	
HCV + ALD	30 (11)	10 (17)	
ALD	20 (8)	7 (13)	
Other	21 (8)	5 (9)	
Child Pugh, *n* (%)			0.361
A	175 (68)	42 (74)	
B	84 (32)	15 (26)	
History of Decompensation, *n* (%)	78 (30)	11 (19)	0.090
Sodium mM, median (IQR)	139 (137–141)	139 (136–141)	0.386
Creatinine mg/dL, median (IQR)	0.9 (0.8–1.1)	0.9 (0.8–1.1)	0.803
Bilirubin mg/dL, median (IQR)	1 (0.6–1.6)	0.8 (0.6–1.3)	0.017
Albumin g/dL, median (IQR)	3.4 (3–3.7)	3.5 (3.1–3.8)	0.166
Albumin g/dL, *n* (%)			0.207
Albumin ≤ 3.4	142 (55)	26 (46)	
Albumin > 3.4	117 (45)	31 (54)	
INR, median (IQR)	1.1 (1–1.2)	1.1 (1–1.2)	0.414
MELD-Na, median (IQR)	9 (7–11)	8 (7–11)	0.077
MELD 3.0, median (IQR)	10 (8–13)	9 (7–12)	0.250
HCC Baseline			
Multifocal, *n* (%)	53 (20)	7 (12)	0.137
Index lesion cm, median (IQR)	2.8 (2.2–3.6)	2.7 (1.9–3.6)	0.118
Index lesion cm, *n* (%)			0.499
Small (<3 cm)	158 (61)	32 (56)	
Intermediate/Large (>3 cm)	101 (39)	25 (44)	
Milan Criteria, *n* (%)	226 (87)	51 (89)	0.639
BCLC Stage, *n* (%)			0.222
A	239 (92)	55 (96)	
B	20 (8)	2 (4)	
ECOG, *n* (%)			0.211
0	164 (63)	41 (72)	
1	95 (37)	16 (28)	
AFP ng/mL, median (IQR)	13 (5.2–59.3)	20.5 (5.3–188.3)	0.170
AFP ng/mL, *n* (%)			0.032
AFP < 20 ng/mL, *n* (%)	154 (60)	29 (50)	
AFP > 20 ng/mL, *n* (%)	104 (40)	28 (50)	
Index Liver-Directed Therapy			
Modality, *n* (%)			0.126
DEE-TACE	109 (42)	16 (28)	
^90^Y	93 (36)	24 (42)	
MWA	57 (22)	17 (30)	
Follow up time from LDT to 1′ Follow up, median (IQR)	40 (32–72)	45 (34–82)	0.153
First Cycle Response, *n* (%)			0.135
ORR	178 (74)	34 (64)	
NOR	61 (26)	19 (36)	

Abbreviations: Interquartile range (IQR), Hepatocellular carcinoma (HCC), Hepatitis C virus (HCV), Nonalcoholic steatohepatitis (NASH), Alcoholic liver disease (ALD), International normalized ratio (INR), Model for end-stage liver disease (MELD); Barcelona Clinic Liver Cancer (BCLC), Eastern Cooperative Oncology Group (ECOG), Alpha-fetoprotein (AFP), Doxorubicin eluting embolic transarterial chemoembolization (DEE-TACE), Yttrium-90 (^90^Y), Microwave ablation (MWA), Liver-directed therapy (LDT), Objective response rate (ORR), Non-objective response rate (NOR).

**Table 6 cancers-16-00212-t006:** Time-to-progression multivariate analysis with transplant evaluation status.

	Univariate		Milan Multivariate Model	
Parameters	*p*-Value	HR (95% CI)	*p*-Value	HR (95% CI)
HCC Imaging No-Show Rate/Month	<0.001	11 (4.8–23.5)	<0.001	5.6 (2.6–11.6)
HCC Imaging Delay/Month	<0.001	1.8 (1.5–2.2)	<0.001	1.7 (1.4–2.1)
HCC Laboratory Delay/Month	<0.001	1.6 (1.4–1.8)	0.169	
HCC Office Delay/Month	<0.001	1.8 (1.6–2.2)	0.683	
Transplant Evaluation Status	0.231		0.822	
Albumin, g/dL				
≤3.4 vs. >3.4	0.041	1.6 (1.0–2.5)	0.028	1.7 (1.0–2.7)
AFP, ng/mL				
>20 vs. <20	<0.001	3.0 (1.9–4.9)	<0.001	2.3 (1.4–3.7)
Milan Criteria				
Outside vs. Within	0.002	2.5 (1.5–4.3)	0.261	
First Cycle Response				
NOR vs. ORR	<0.001	5.8 (3.7–9.2)	<0.001	4.1 (2.6–6.7)

Abbreviations: Hazard ratio (HR, Confidence Interval (CI), Hepatocellular carcinoma (HCC), Alpha-fetoprotein (AFP), Objective response rate (ORR), Non-objective response rate (NOR).

## Data Availability

The dataset generated for the current study is available from the corresponding author upon reasonable request.

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
