# Peer review of "Imaging Delay Following Liver-Directed Therapy Increases Progression Risk in Early- to Intermediate-Stage Hepatocellular Carcinoma"

_cancers, 2024, doi:10.3390/cancers16010212_

Round 1
Reviewer 1 Report
Comments and Suggestions for Authors
The impact of HCC care delay on HCC progression among patients on the liver transplantation waiting list is an important issue. Most of the article is well written. However, some points should be solved:
[major problem]
1. the univariate analysis should be expressed in the Tables 4 and 6
2. Please show a reference to support the cut-off value of 50 for AFP (Tables 4, 5, and 6) (the usual cut-off is 10 or 400 ng/mL)
[minor problem]
1. The abbreviation of "LDT" should be shown first (line 52).
2. What is the meaning of p-value in Table 1?
3. Table legends should not be in the Tables (Tables 2 and 3)
4. The P-values are usually placed on the right side of HR (Tables 4 & 6)
5. Multiple references should be put in the same brackets (lines 277, 279, 282, 290, 291, 342, 343)
Author Response
Reviewer #1
- the univariate analysis should be expressed in the Tables 4 and 6
- In the revised manuscript, Tables 4 and 6 were amended to include both the univariate and multivariate outcomes for variables of interest.
- Please show a reference to support the cut-off value of 50 for AFP (Tables 4, 5, and 6) (the usual cut-off is 10 or 400 ng/mL)
- In the revised manuscript, cut-off value for AFP has been adjusted to 20 ng/mL based on data from our prior publications[1,2]. Analysis for the manuscript has been updated.
- The abbreviation of "LDT" should be shown first (line 52).
- LDT is defined at first usage in the revision.
- What is the meaning of p-value in Table 1?
- We apologize for the confusion. We believe the Table title did not accurately reflect the content of the Table. The cohort demographics were placed in the data column and evaluated for univariate TTP in the P-value column. To avoid confusion, Table 1 has been retitled to reflect both components and the manuscript text revised at both callouts to focus the reader to the specific column of interest. Additionally, we have added an additional column between the summary statistic and univariate P-value to define the univariate input (dichotomous grouping vs. per unit value).
- We apologize for the confusion. We believe the Table title did not accurately reflect the content of the Table. The cohort demographics were placed in the data column and evaluated for univariate TTP in the P-value column. To avoid confusion, Table 1 has been retitled to reflect both components and the manuscript text revised at both callouts to focus the reader to the specific column of interest. Additionally, we have added an additional column between the summary statistic and univariate P-value to define the univariate input (dichotomous grouping vs. per unit value).
- Table legends should not be in the Tables (Tables 2 and 3)
- We apologize for the confusion, but it is unclear why Tables 2 and 3 should specifically not have table legends. If the issue is related to placement in document, we will work with the editorial staff to ensure table legends are placed in the correct location.
- The P-values are usually placed on the right side of HR (Tables 4 & 6)
- The authors respectfully disagree based on our prior experience, but ultimately defer to the editorial staff for preferred placement and agree to revise accordingly.
- Multiple references should be put in the same brackets (lines 277, 279, 282, 290, 291, 342, 343)
- We apologize for the error and have corrected several reference content and formatting issues in the revision.
References
- Nunez, K.G.; Sandow, T.; Fort, D.; Patel, J.; Hibino, M.; Carmody, I.; Cohen, A.J.; Thevenot, P. Baseline Alpha-Fetoprotein, Alpha-Fetoprotein-L3, and Des-Gamma-Carboxy Prothrombin Biomarker Status in Bridge to Liver Transplant Outcomes for Hepatocellular Carcinoma. Cancers (Basel) 2021, 13, doi:10.3390/cancers13194765.
- Nunez, K.G.; Sandow, T.; Patel, J.; Hibino, M.; Fort, D.; Cohen, A.J.; Thevenot, P. Hypoalbuminemia Is a Hepatocellular Carcinoma Independent Risk Factor for Tumor Progression in Low-Risk Bridge to Transplant Candidates. Cancers (Basel) 2022, 14, doi:10.3390/cancers14071684.
Reviewer 2 Report
Comments and Suggestions for Authors
The manuscript demonstrates the burden of delay in HCC management on oncological outcomes.
The rate of no-show appointments affects the risk of oncological worsening of patients after liver-directed therapies (LDT).
The work is well written and demonstrates a problem in all Centers.
However, there are some things to consider:
- the bibliography needs to be reviewed as it is not in order (citation no. 1 on the second page)
- The abbreviations need to be reviewed (the abbreviation for liver-directed therapies is missing at the beginning)
- the statistics need to be reviewed: In Table 1, the comparisons between the quantitative variables are not clear
Ultimately, the work highlights a significant problem in managing outpatient follow-up and referral to transplant centers.
However, the scientific way to demonstrate organizational distress can be developed in a more effective and smart way.
Comments on the Quality of English Language
The English language is well defined and written
Author Response
Reviewer #2
- the bibliography needs to be reviewed as it is not in order (citation no. 1 on the second page)
- We apologize for the error and have corrected several reference content and formatting issues in the revision.
- The abbreviations need to be reviewed (the abbreviation for liver-directed therapies is missing at the beginning)
- LDT is defined at first usage in the revision.
- the statistics need to be reviewed: In Table 1, the comparisons between the quantitative variables are not clear
- We apologize for the confusion. We attempted to consolidate the cohort demographics and disease baseline overview with the univariate analysis of TTP for those factors. To improve clarity, we have added an additional column between the summary statistic and univariate P-value to define the univariate input (dichotomous grouping vs. per unit value).
Reviewer 3 Report
Comments and Suggestions for Authors
Thanks for allowing me to review this paper. In this study, authors suggested that HCC imaging delay is associated with TTP in bridge/downstage to LT or definitive treatment in nonresectable HCC following LDT. It is an excellent work and well-written. However, the authors need to provide some information to improve this manuscript.
1. Please start with the primary finding of this study in the discussion part. Primary findings are missing in the first paragraph.
2. Please provide the clinical implication of this study in the discussion part.
3. Please extend the conclusion section.
Comments on the Quality of English Language
Minor revision is needed.
Author Response
Reviewer #3
- Please start with the primary finding of this study in the discussion part. Primary findings are missing in the first paragraph.
- The first paragraph of the discussion has been revised to state the primary findings.
- Please provide the clinical implication of this study in the discussion part.
- We apologize for this oversight and have elaborated on the clinical implications of the study in the revised discussion.
- Please extend the conclusion section.
- We have expanded the concluding paragraph to the manuscript in the revision.
Round 2
Reviewer 1 Report
Comments and Suggestions for Authors
The authors replied to my questions adequately.
Author Response
Thank you for your review.
Reviewer 2 Report
Comments and Suggestions for Authors
The authors highlighted the role of the socio-economic level of HCC patients in the follow-up delay and, consequently, on the oncological outcome.
The work has been improved from a design and data management point of view.
It could be further enhanced in significance by comparing the cohort broken down by socio-economic factors.
Statistical tests performed "per unit value" continue to be unclear.
Comments on the Quality of English Language
The English language was well optimized.
Author Response
Reviewer #2
- the bibliography needs to be reviewed as it is not in order (citation no. 1 on the second page)
- We apologize for the error and have corrected several reference content and formatting issues in the revision.
- The abbreviations need to be reviewed (the abbreviation for liver-directed therapies is missing at the beginning)
- LDT is defined at first usage in the revision.
- the statistics need to be reviewed: In Table 1, the comparisons between the quantitative variables are not clear
- We apologize for the confusion. We attempted to consolidate the cohort demographics and disease baseline overview with the univariate analysis of TTP for those factors. To improve clarity, we have added an additional column between the summary statistic and univariate P-value to define the univariate input (dichotomous grouping vs. per unit value).
- The authors highlighted the role of the socio-economic level of HCC patients in the follow-up delay and, consequently, on the oncological outcome.
- Completed
- The work has been improved from a design and data management point of view.
- Completed
- It could be further enhanced in significance by comparing the cohort broken down by socio-economic factors.
- We have investigated the Area of Deprivation Index (ADI) both at the national and state level. ADI was available in 65% (206/316) of cohort. Due to the local area of Louisiana being so depressed compared to the national ranking, we have included the ADI state rank to better illustrate local variation, however, no trend emerged (see figure below). We are currently focused on identifying the specific exacerbating factor(s) in the dataset to provide a more granular assessment of the socioeconomic stressors contributing to HCC care delay.
- Statistical tests performed "per unit value" continue to be unclear.
- We apologize for the additional confusion. We have added in each unit change for those parameters in Table 1.
